# Spatial patterning of liver progenitor cell differentiation mediated by cellular contractility and Notch signaling

Kerim B Kaylan[1], Ian C Berg[1], Matthew J Biehl[2], Aidan Brougham-Cook[1], Ishita Jain[1], Sameed M Jamil, Lauren H Sargeant, Nicholas J Cornell, Lori T Raetzman[2], Gregory H Underhill*

[1]Department of Bioengineering, University of Illinois at Urbana-Champaign, Urbana, United States; [2]Department of Molecular and Integrative Physiology, University of Illinois at Urbana-Champaign, Urbana, United States

**Abstract** The progenitor cells of the developing liver can differentiate toward both hepatocyte and biliary cell fates. In addition to the established roles of TGFβ and Notch signaling in this fate specification process, there is increasing evidence that liver progenitors are sensitive to mechanical cues. Here, we utilized microarrayed patterns to provide a controlled biochemical and biomechanical microenvironment for mouse liver progenitor cell differentiation. In these defined circular geometries, we observed biliary differentiation at the periphery and hepatocytic differentiation in the center. Parallel measurements obtained by traction force microscopy showed substantial stresses at the periphery, coincident with maximal biliary differentiation. We investigated the impact of downstream signaling, showing that peripheral biliary differentiation is dependent not only on Notch and TGFβ but also E-cadherin, myosin-mediated cell contractility, and ERK. We have therefore identified distinct combinations of microenvironmental cues which guide fate specification of mouse liver progenitors toward both hepatocyte and biliary fates.
DOI: https://doi.org/10.7554/eLife.38536.001

*For correspondence:
gunderhi@illinois.edu

Competing interests: The authors declare that no competing interests exist.

## Introduction

The cells which populate the hepatic diverticulum during development and later serve as the source of liver parenchyma are termed bipotential progenitor cells, or hepatoblasts, as they are capable of differentiating toward both hepatocytic and biliary epithelial cell fates. While differentiation of liver progenitors toward a hepatocytic fate is guided chiefly by signaling through Wnt, HGF, and FGF (*Micsenyi et al., 2004*; *Berg et al., 2007*; *Schmidt et al., 1995*), biliary fate is regulated by Notch and TGFβ signaling (*Kodama et al., 2004*; *Clotman et al., 2005*; *Zong et al., 2009*). Specifically, a gradient of TGFβ activity caused in part by expression of TGFβR2 and TGFβR3 in the periportal region leads to differentiation of progenitors toward a biliary epithelial fate (*Clotman et al., 2005*). In patients with Alagille syndrome, mutations in the ligand *JAG1* or receptor *NOTCH2* are associated with bile duct paucity and cholestasis (*Li et al., 1997*; *Oda et al., 1997*; *McDaniell et al., 2006*). Zong *et al.* further underlined the importance of Notch in particular for both biliary cell fate and morphogenesis by showing that deletion of the Notch effector *Rbpj* results in reduction of both biliary fate and abnormal tubulogenesis (*Zong et al., 2009*).

Thus, the progenitor cells of the developing liver integrate a diverse set of biochemical cues during fate specification. Several recent lines of evidence suggest, however, that liver progenitor cells are influenced not only by biochemical cues but also biophysical parameters in their microenvironment. Using combinatorial extracellular matrix (ECM) protein arrays, we showed that TGFβ-induced biliary differentiation of liver progenitor cells is coordinated by both substrate stiffness and matrix

**eLife digest** Children are said to be a product of both nature and nurture – of their genes and the environment in which they are raised. The cells of the growing liver are not so different in this sense. As the liver of a fetus develops, immature cells called liver progenitors mature to become one of two types of adult cells: the hepatocytes that form the bulk of the liver, or the biliary cells that make up the bile duct. The traditional view is that genetic factors mainly control which cell type the progenitor cells become. However, recent research suggests that the environment around the cells matters more in this process than once thought.

Cells can respond to the physical properties of their environment, such as the structure and stiffness of the surrounding tissue. These properties change as the liver develops, and can also be altered by disease. For example, damaged liver cells can spit out proteins that harden and form stiff scars. This raises a question: do changes in stiffness affect how progenitor cells behave?

To answer this question, Kaylan et al. printed collagen in circular patterns and grew liver progenitor cells on them. The cells at the edges of the circular patterns matured into bile duct cells, while those in the center became hepatocytes. The stiffness felt by the cells was then determined by measuring the level of mechanical stress that they experienced. This revealed that the cells at the edge of the collagen pattern – the cells that became bile duct cells – were under most stress. In addition, more bile duct cells formed when progenitor cells were grown on a stiffer collagen pattern.

Overall, the results reported by Kaylan et al. suggest that the stiffness of the environment, and the resulting stresses on a progenitor cell, can influence how it matures. As well as helping us to understand how the liver develops, this knowledge could also help us to treat a group of diseases called cholangiopathies, in which the bile ducts become inflamed. These diseases are thought to be caused by certain cells (which are similar to liver progenitor cells) maturing to become incorrect cell types. Future studies could determine if preventing changes in stiffness in the environment of these cells, or slowing their response to such changes, would help patients.

DOI: https://doi.org/10.7554/eLife.38536.002

context and is further correlated with cell contractility (*Kourouklis et al., 2016*). Several groups have established mechanosensing *via* the transcriptional co-activator YAP and further elaborated a novel role for this protein in the developing cells of the liver (*Camargo et al., 2007*; *Dupont et al., 2011*; *Yimlamai et al., 2014*; *Lee et al., 2016*). This is particularly interesting in the context of liver progenitor fate specification because YAP has been shown to regulate both Notch signaling and TGFβ in liver cells (*Yimlamai et al., 2014*; *Lee et al., 2016*). However, the potential link between mechanical sensing and the fate specification of liver progenitor cells has yet to be fully defined.

Here, we utilize microarrayed patterns of ECM co-printed with Notch ligands to provide a controlled biochemical and biomechanical environment for liver progenitor cell differentiation. We characterize spatially-localized, segregated differentiation of these progenitor cells toward biliary fates at the periphery of patterns and hepatocytic fates near the center of patterns. We employ traction force microscopy (TFM) to measure cell-generated forces, observing high stresses coincident with peripheral biliary differentiation. Further, we explore the dependence of peripheral biliary differentiation of progenitors on mechanotransduction pathway activity and expression of the Notch ligands *Jag1* and *Dll1*. Collectively, our findings provide support for a model of liver progenitor differentiation which includes mechanical signaling as a key regulator of spatially-segregated progenitor differentiation and downstream biliary morphogenesis.

## Results

### Liver progenitor fate segregation in arrayed patterns is dependent on Notch signaling and substrate stiffness

We have previously observed peripheral expression of the biliary marker osteopontin (OPN) in liver progenitors on arrayed patterns containing both ECM proteins and Notch ligands (*Kaylan et al., 2016*). In order to better characterize the expression profile of cells in the periphery *vs.* center, we

fabricated arrays of circular patterns (~600 µm diameter) containing the ECM protein collagen I paired with either control IgG or Fc-recombinant Notch ligands (DLL1, DLL4, and JAG1). These ligands were pre-conjugated to Protein A/G so as to increase ligand functionality by clustering and orientation. Bipotential mouse embryonic liver (BMEL) progenitor cells, which are capable of assuming a hepatocytic or biliary fate (*Strick-Marchand and Weiss, 2002*), were seeded on these Notch ligand arrays and cultured under differentiation conditions for $t = 72\,\mathrm{h}$, at which point we immunolabeled for OPN and the hepatocytic marker albumin (ALB). Within these defined multicellular geometries, we observed OPN$^+$ cells at the periphery of patterns while ALB$^+$ cells were located centrally (*Figure 1A*). Counts of cells that were OPN$^+$ peaked at the periphery and increased with the presentation of Notch ligands, particularly DLL4 (*Figure 1B*). However, counts of cells that were ALB$^+$ cells indicated central localization and only moderate induction by ligand in the center of patterns (*Figure 1C*). Multiple regression analysis of these data generated coefficient estimates (β) for each presented ligand, corresponding to the mean change in cell counts from control IgG (*Figure 1E* and *Figure 1F*). These coefficient estimates confirmed increases in both peripheral OPN$^+$($\beta = 37.5$, $P < 0.001$) and central ALB$^+$ ($\beta = 5.64$, $P < 0.001$) cell counts upon presentation with DLL4. Evaluation of the expression of the biliary transcription factor SOX9 and hepatocytic transcription factor HNF4A revealed segregation similar to that of OPN and ALB (*Figure 1D*). Specifically, SOX9-expressing cells were at the periphery while HNF4A-expressing cells were central. We also evaluated expression of the biliary marker cytokeratin 19 (CK19) (*Figure 1—figure supplement 1*) and observed 1.6 times greater intensity in cells at the periphery compared to those in the center ($P < 0.001$) (*Figure 1—figure supplement 1*). We observed peripheral expression of both OPN and CK19 at $t = 24\,\mathrm{h}$, suggesting that segregation starts earlier than $t = 72\,\mathrm{h}$ and is less likely to be dependent on cell motility mechanisms (*Figure 1—figure supplement 2A*). Measurements of cell density across the island at $t = 72\,\mathrm{h}$ indicated uniform density with radius, ruling out cell condensation as a mechanism of differentiation (*Figure 1—figure supplement 2B*). In preliminary experiments, we determined that patterns of approximately 600 µm diameter would lead to consistent patterned differentiation. Accordingly, for most our studies here, 600 µm diameter patterns were utilized. However, to examine potential effects of pattern diameter, we generated complementary array sets that resulted in cellular island diameters of 300 µm and 1000 µm, in addition to 600 µm (*Figure 1—figure supplement 3A*). Quantification of peak OPN$^+$ cell counts on these pattern sizes indicated that biliary differentiation remained confined to the periphery independent of pattern size (*Figure 1—figure supplement 3B*). Together, these data establish spatially-segregated liver progenitor fates in arrayed patterns with central hepatocytic differentiation and peripheral biliary differentiation.

We next asked if Notch signaling is necessary for peripheral biliary differentiation in arrayed patterns. We treated cultures with an inhibitor of Notch signaling (γ-secretase inhibitor X, GSI) and observed reduction in OPN$^+$ cell counts at the periphery (*Figure 2A and B*). Prompted by previous experiments which showed that liver progenitor differentiation is sensitive to substrate stiffness (*Kourouklis et al., 2016*), we also evaluated progenitor differentiation on soft (4 kPa) rather than stiff (30 kPa) substrates, observing decreased counts of peripheral OPN$^+$ cells and similar responsiveness to GSI (*Figure 2A and B*, *Figure 2—figure supplement 1*). Multiple regression analysis of these data confirmed reduction in peripheral OPN$^+$ with both GSI treatment ($\beta = -9.99$, $P < 0.001$) and culture on 4 kPA substrates ($\beta = -3.10$, $P = 0.00292$) (*Figure 2—figure supplement 2*). ALB$^+$ cell counts increased with GSI treatment ($\beta = 5.41$, $P < 0.001$), suggesting that hepatocytic differentiation is inhibited by active Notch signaling (*Figure 2—figure supplement 3* and *Figure 2—figure supplement 2*). We also evaluated expression of SOX9 and HNF4A, observing reduction in peripheral SOX9 expression and an increase in central HNF4A expression on soft substrates compared to stiff (*Figure 2C*). Quantification of immunolabel intensity for SOX9 and HNF4A on both soft and stiff substrates confirmed our qualitative observations (*Figure 2D*), indicating a 74.7% increase in overall SOX9 intensity on 30 kPa substrates (relative to 4 kPa substrates, $P < 0.001$) and 40.6% increase in overall HNF4A intensity on 4 kPa substrates (relative to 30 kPa substrates, $P < 0.001$). Using in situ hybridization of mRNA, we characterized the expression of Notch family members in arrayed patterns of liver progenitors. To do so, we validated several probes against *Jag1*, *Dll1*, and *Notch2* (data not shown). When used to detect mRNA in arrayed patterns fabricated on stiff (30 kPa) substrates, we observed peripheral localization of *Jag1*, *Dll1*, and *Notch2* (*Figure 2E*). Presentation of the ligand DLL4 induced rearrangement of this expression pattern, specifically causing an increase in centrally-located cells expressing mRNA for each gene. On soft (4 kPa) substrates, we observed

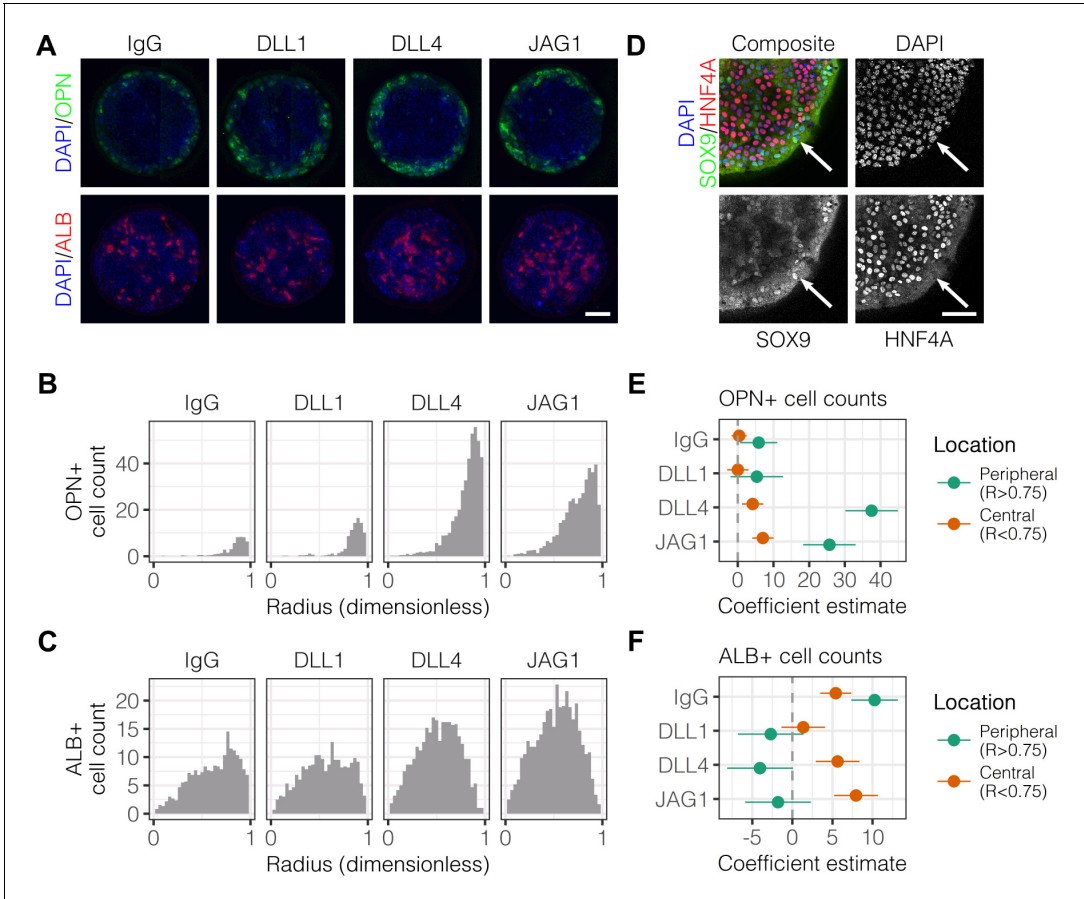

**Figure 1.** Localized differentiation of liver progenitors in arrayed patterns. (A) Immunolabeling of BMEL cells for the biliary marker OPN and hepatocyte marker ALB on arrayed collagen I patterns with control IgG or Fc-recombinant Notch ligands DLL1, DLL4, and JAG1. (B) Quantification of OPN[+] cell counts as a function of radial distance from the centroid of each island. (C) Quantification of ALB[+] cell counts as a function of radial distance from the centroid of each island. (D) Immunolabeling of BMEL cells presented with DLL4 for the biliary transcription factor SOX9 and hepatocyte transcription factor HNF4A. Arrow in each image indicates the same SOX9+/HNF4A− cell. Scale bar indicates 75 μm. (E, F) Regression analysis of OPN[+] and ALB + cell counts. Data in *Figure 1B* and *Figure 1C* were separated into peripheral and central subsets for which dimensionless radius was greater than 0.75 ($R > 0.75$) and less than 0.75 ($R < 0.75$). Separate multiple regression models were generated for each data subset for which coefficient estimates (corresponding to mean change in cell counts) and 95% CI were plotted for OPN+ (E) and ALB[+] (F) cells. For each factor, 95% CI that do not intersect with the dashed line indicate regression coefficient estimates for which $P < 0.05$. (A, E) Scale bars indicate 150 μm.

DOI: https://doi.org/10.7554/eLife.38536.003

The following source data and figure supplements are available for figure 1:

**Source data 1.** Summary table for OPN data in *Figure 1B*.
DOI: https://doi.org/10.7554/eLife.38536.007

**Source data 2.** Summary table for ALB data in *Figure 1C*.
DOI: https://doi.org/10.7554/eLife.38536.008

**Figure supplement 1.** Immunolabeling and quantification of CK19.
DOI: https://doi.org/10.7554/eLife.38536.004

**Figure supplement 2.** Immunolabeling of OPN and CK19 at $t = 24\,\mathrm{h}$ and cell density with radius at $t = 72\,\mathrm{h}$.
DOI: https://doi.org/10.7554/eLife.38536.005

**Figure supplement 3.** Immunolabeling for OPN with 300, 600, and 1000 μm diameter patterns.
DOI: https://doi.org/10.7554/eLife.38536.006

similar mRNA expression for cells presented with IgG but no longer observed ligand-induced central expression for *Jag1* and *Notch2*. This loss of ligand-induced central expression on soft substrates suggests that the responsiveness of liver progenitors to Notch ligand is enhanced by stiffer substrates. Collectively, these data show that segregation of liver progenitor fates is dependent on both Notch signaling and substrate stiffness.

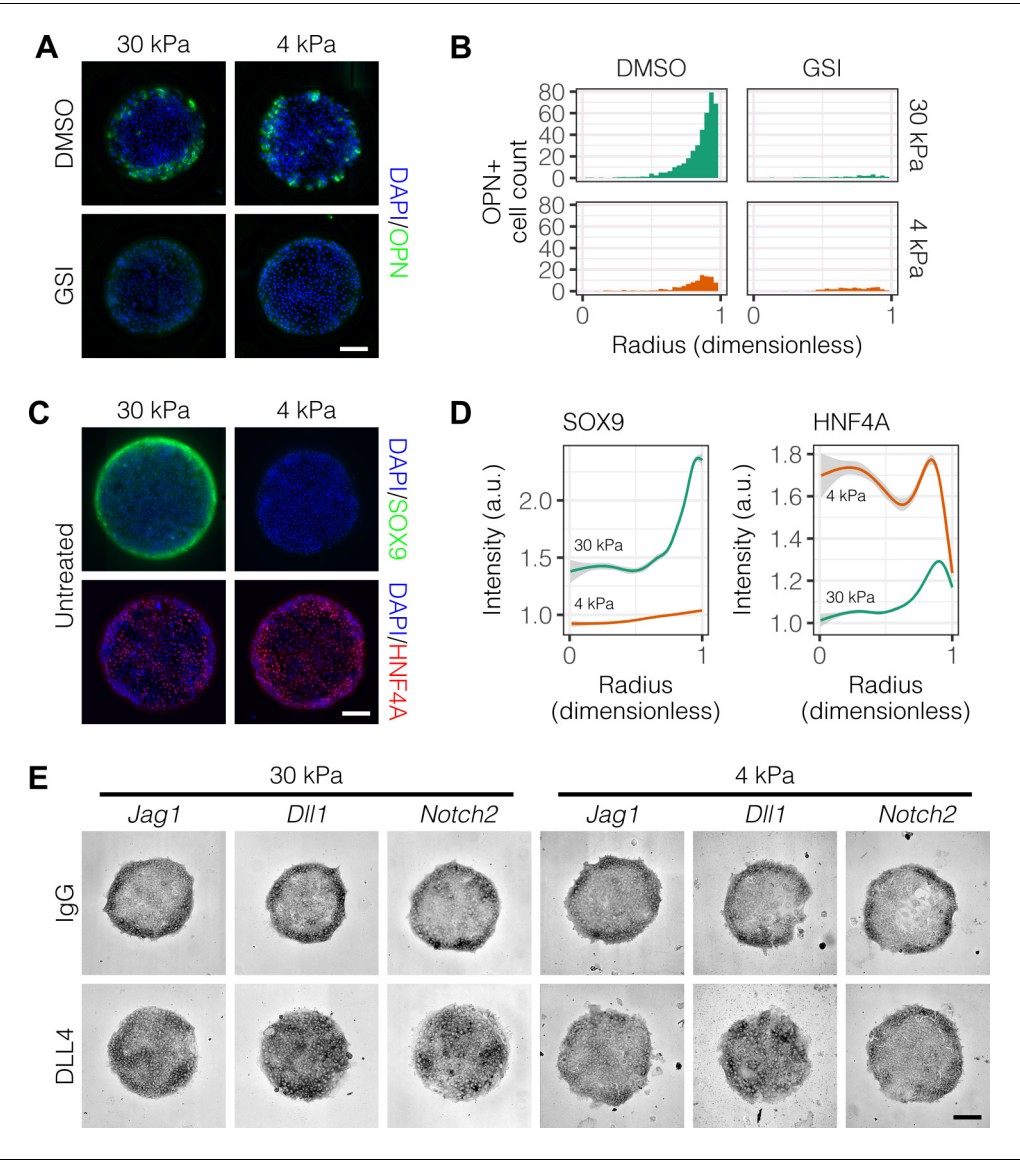

**Figure 2.** Peripheral biliary differentiation is dependent on both Notch signaling and substrate stiffness. (**A**) Immunolabeling for OPN of BMEL cells presented with DLL4 on 30 kPa and 4 kPa substrates. Cells were treated with vehicle control (DMSO) or an inhibitor of Notch signaling (γ-secretase inhibitor X, GSI, 5 μM). (**B**) Quantification of OPN$^+$ cell counts on 30 kPa and 4 kPa substrates after treatment with DMSO or GSI. (**C**) Immunolabeling for SOX9 and HNF4A of BMEL cells on 30 kPa and 4 kPa substrates. (**D**) Quantification of SOX9 and HNF4A intensity on 30 kPa and 4 kPa substrates. (**E**) RNA in situ hybridization for *Jag1*, *Dll1*, and *Notch2* on 30 kPa and 4 kPa substrates. Cells were exogenously presented with IgG or DLL4. (**A, C, E**) Scale bars indicate 150 μm. (**B, D**) Mean ± 95% CI.

DOI: https://doi.org/10.7554/eLife.38536.009

The following source data and figure supplements are available for figure 2:

**Source data 1.** Summary table for OPN data in *Figure 2B* and *Figure 2—figure supplement 1*.
DOI: https://doi.org/10.7554/eLife.38536.013
**Source data 2.** Summary table for SOX9 and HNF4A data in *Figure 2D*.
DOI: https://doi.org/10.7554/eLife.38536.014
**Figure supplement 1.** Quantification of OPN$^+$ cell counts in arrayed patterns.
DOI: https://doi.org/10.7554/eLife.38536.010
**Figure supplement 2.** Regression analysis of OPN+ and ALB+ cell counts.
DOI: https://doi.org/10.7554/eLife.38536.011
**Figure supplement 3.** Quantification of ALB$^+$ cell counts in arrayed patterns.

*Figure 2 continued on next page*

*Figure 2 continued*

DOI: https://doi.org/10.7554/eLife.38536.012

## TGFβ and E-cadherin have distinct roles in fate segregation

Previous studies have delineated a role for TGFβ in liver progenitor differentiation (*Clotman et al., 2005*), and we have described interactions between TGFβ and Notch signaling in this context (*Kaylan et al., 2016*). To determine if TGFβ is involved in the generation of biliary cells at the periphery of the arrayed patterns, we treated cells with an inhibitor of TGFβ type I receptor kinase signaling (SB-431542) or stimulated with exogenous TGFβ1 (*Figure 3A*). Treatment with SB-431542 reduced the peripheral count of OPN+ cells while increasing central expression of HNF4A (*Figure 3A*, *Figure 3—figure supplement 1*). In contrast, treatment with TGFβ1 increased counts of OPN+ cells uniformly across the patterns irrespective of ligand presented (*Figure 3A and B*), in agreement with previous efforts showing uniform induction of OPN on patterns of smaller diameter (150 μm) (*Kourouklis et al., 2016*). Similarly, in situ hybridization for *Jag1*, *Dll1*, and *Notch2* mRNA showed uniform induction across the patterns with TGFβ1 treatment (*Figure 3—figure supplement*

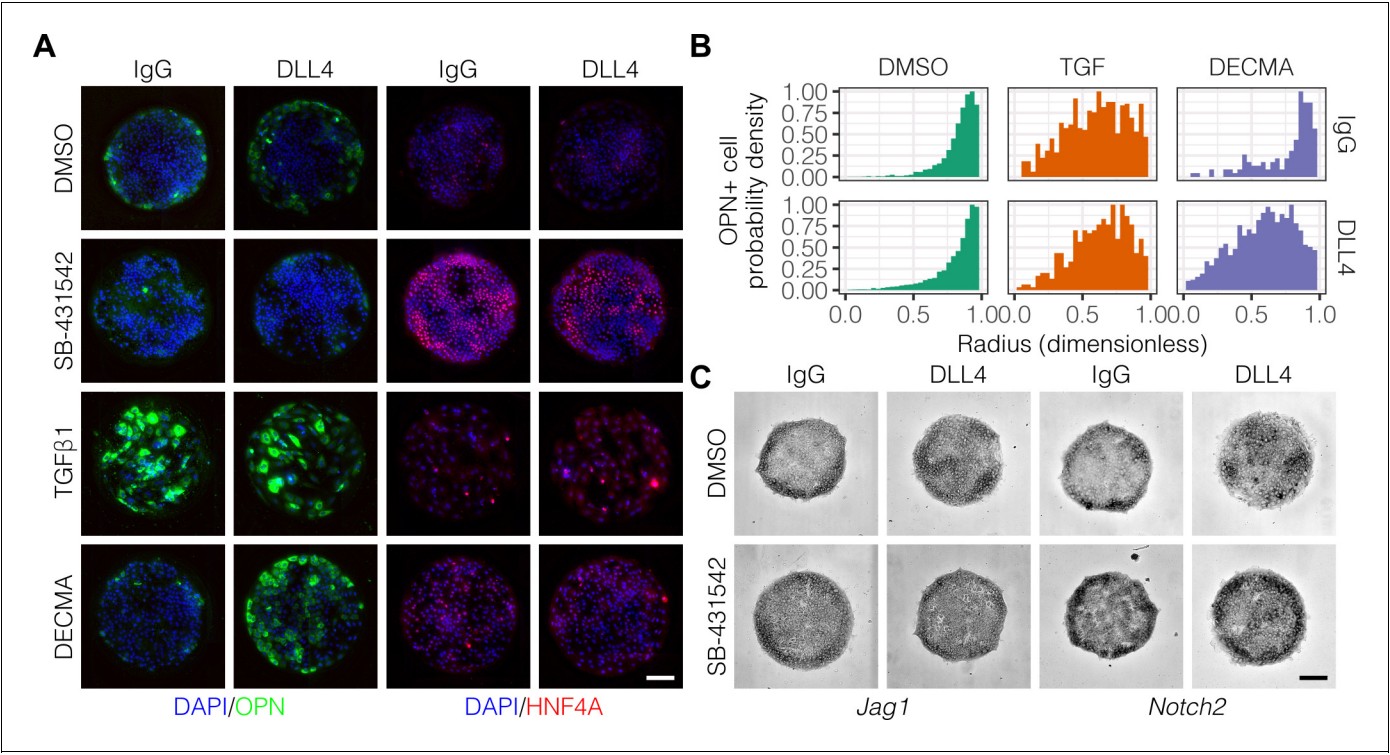

**Figure 3.** TGFβ signaling and cell–cell interaction strength modulate pattern formation. (**A**) Immunolabeling for OPN and HNF4A of BMEL cells presented with IgG and DLL4 on 30 kPa substrates. Cultures were treated with vehicle control (DMSO), inhibitor of TGFβ signaling (SB-431542, 10 μm), TGFβ1 (1.5 ng/ml), or functional anti-E-cadherin (DECMA, 10 μg/ml). (**B**) Quantification of OPN+ cell probability density distributions after treatment with DMSO, TGFβ1, or DECMA. (**C**) RNA in situ hybridization for *Jag1* and *Notch2* of cells exogenously presented with IgG or DLL4 and treated with DMSO or SB-431542. (**A**, **C**) Scale bars indicate 150 μm.

DOI: https://doi.org/10.7554/eLife.38536.026

The following source data and figure supplements are available for figure 3:

**Source data 1.** Summary table for OPN+ probability density data in *Figure 3B*.
DOI: https://doi.org/10.7554/eLife.38536.029

**Figure supplement 1.** Quantification of OPN+ cell counts and HNF4A intensity with SB-431542 and DECMA treatment.
DOI: https://doi.org/10.7554/eLife.38536.027

**Figure supplement 2.** TGFβ1 induces Notch ligand and receptor expression uniformly.
DOI: https://doi.org/10.7554/eLife.38536.028

*2*). Interestingly, we observed loss of cell–cell junctional interactions in cells treated with TGFβ1, which is thought to be a consequence of inhibition of E-cadherin expression by the Snail family of transcription factors (*Cano et al., 2000*; *Vincent et al., 2009*). To ascertain the impact of E-cadherin function without activation of the other regulatory programs of TGFβ, we treated cells with a functional antibody against E-cadherin (DECMA) (*Figure 3A*). In contrast with our observations following treatment with TGFβ1, we observed differential responsiveness to control IgG and DLL4 presentation (*Figure 3B*). Specifically, presentation of DLL4 to cells treated with DECMA resulted in uniform induction of OPN$^+$ cells across the patterns. We confirmed this observation using the Kolmogorov–Smirnov test, which showed that the difference between the IgG and DLL4 probability density distributions (measured by $D$, the supremum distance) was greater for DECMA ($D = 0.437$, $P < 0.001$) compared to both DMSO ($D = 0.0655$, $P < 0.001$) and TGFβ1 ($D = 0.0848$, $P = 0.0350$). Last, although inhibition of TGFβ by treatment with SB-431542 reduced OPN$^+$ cell counts, mRNA in situ hybridization of cultures treated with SB-431542 indicated that both *Jag1* and *Notch2* remain expressed at the periphery (*Figure 3C*). However, SB-4315412 treatment reduced expression of both *Jag1* and *Notch2* in centrally-located cells presented with DLL4 (*Figure 3C*), which we had previously observed in untreated cultures (*Figure 2E*). These data therefore demonstrate that TGFβ only partially regulates fate segregation and that these effects are additionally mediated by cell–cell junctional interactions through E-cadherin.

## Simulated and experimental mechanical stresses are coincident with peripheral biliary fate

Others have established a role for mechanical stresses in multicellular pattern formation and stem cell differentiation, specifically observing collection of mechanical stresses at the corners and edges of geometric shapes (*Nelson et al., 2005*; *Ruiz and Chen, 2008*; *Kilian et al., 2010*; *Ma et al., 2015*). Having previously demonstrated a combinatorial role for biochemical and biomechanical stimuli in liver progenitor cell fate (*Kourouklis et al., 2016*), we hypothesized that mechanical stress gradients are involved in the segregation of liver progenitor fates arrayed patterns. To obtain theoretical predictions of mechanical stress, we used finite element modeling (FEM) of an active layer (*i. e.*, the cell monolayer) of 600 μm diameter bound to a passive substrate with fixed lower boundary (*Figure 4A*). We observed peak stresses of 150 Pa at the periphery of the active layer (*Figure 4B*), in agreement with previous simulations (*Nelson et al., 2005*). Next, we used TFM to obtain experimental measurements in liver progenitor cells, observing that traction stresses are collected at the periphery of patterns on both 30 kPa and 4 kPa substrates (*Figure 4C*). The peak magnitude and distribution of stresses across the patterns did not vary with ligand presentation (*Figure 4D*). However, we did observe that central cells ($R < 0.75$) on 30 kPa substrates exerted stresses averaging to 32.9 Pa, which was statistically greater than the 16.2 Pa of stress exerted by cells on 4 kPa substrates ($P < 0.001$). TFM measurements of cells treated with GSI showed that Notch signaling was not upstream of traction stress generation at the periphery (*Figure 4—figure supplement 1*). In contrast, inhibition of TGFβ by treatment with SB-431542 resulted in more uniform traction stress distributions in cells presented with both IgG and DLL4 (*Figure 4—figure supplement 2*). Intriguingly, treatment with functional antibody against E-cadherin (DECMA) resulted in more uniform traction stress distribution in cells presented with IgG but not DLL4, indicating that ligand presentation in the context of reduced cell–cell interactions induces cell-generated traction stresses. In sum, these data show that mechanical stresses are collected at the periphery, coincident with peripheral biliary fate, and are further dependent on TGFβ signaling and E-cadherin interactions between cells.

## Gradients of mechanotransduction pathway activity specify biliary fate

Having established the presence of gradients of mechanical stress in patterns, we next hypothesized that these gradients are involved in the segregated differentiation of liver progenitors. In order to first determine whether regulation by this gradient of mechanical stress is consistent with known modes of Notch signaling, we used a lattice-based computational model described by the groups of Elowitz and Sprinzak (*Sprinzak et al., 2010*; *Formosa-Jordan and Sprinzak, 2014*). We adapted their computational model to include: (1) fixed boundary conditions to better represent the physical boundary of our arrayed patterns; and (2) an additional term representing the effect of the stress gradient on expression of both Notch receptor and ligand, as observed in our mRNA in situ

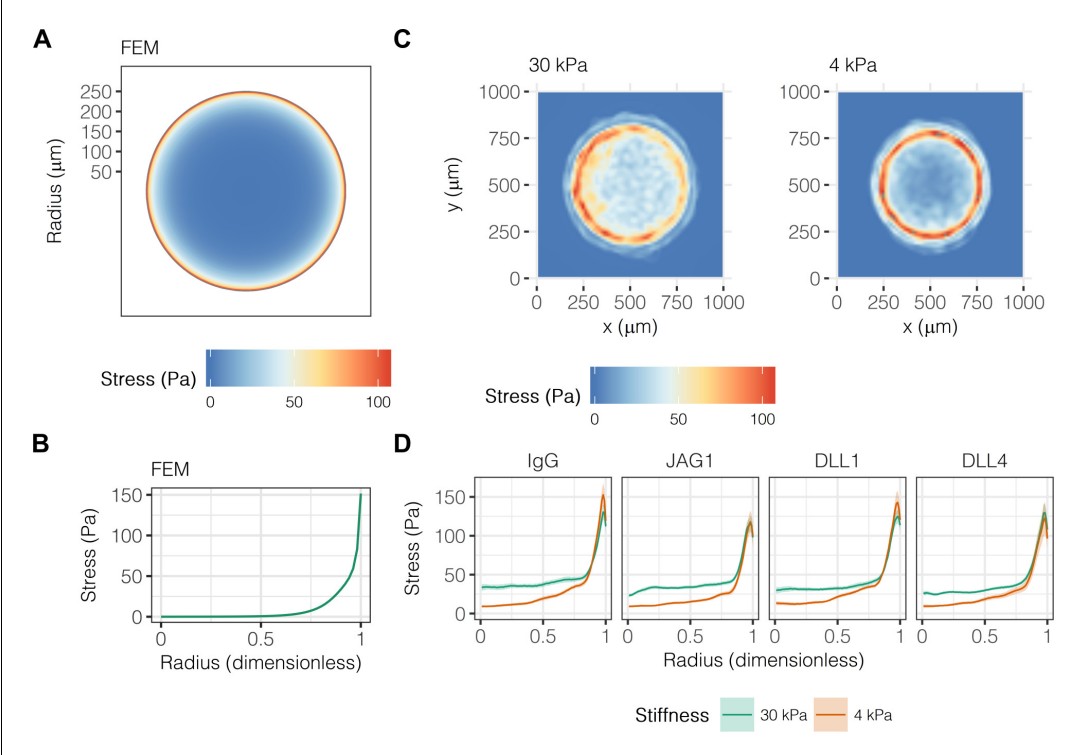

**Figure 4.** Liver progenitors in arrayed patterns generate gradients of traction stress independent of ligand presentation. (**A**) Simulated finite element modeling (FEM) stress profile of arrayed patterns. (**B**) Stress from FEM as a function of radius. (**C**) Experimental stress profiles obtained by traction force microscopy (TFM) of BMEL cells on 30 kPa and 4 kPa substrates presented with DLL4. (**D**) Stress from TFM as a function of radius, substrate stiffness (30 kPa and 4 kPa), and ligand (IgG, JAG1, DLL1, DLL4). Mean ± 95% CI.

DOI: https://doi.org/10.7554/eLife.38536.015

The following figure supplements are available for figure 4:

**Figure supplement 1.** Effect of Notch inhibition on experimental stress profiles.
DOI: https://doi.org/10.7554/eLife.38536.016

**Figure supplement 2.** Effect of TGFβ and E-cadherin inhibition on experimental stress profiles.
DOI: https://doi.org/10.7554/eLife.38536.017

hybridization experiments. A model of *trans*-activation ($K_t = 10$ and $K_c = 0$) with increasing stress gradient strength ($b = 0, 0.5, 5$) produced segregation of fates qualitatively similar to our experimental results (**Figure 5A**). Concentration profiles of Notch receptor and repressor, a measure of Notch signaling activity, in models including *trans*-activation and steeper stress gradients were also qualitatively consistent with our experimental data (**Figure 5B**). Notably, simulations without stress suggested a biphasic distribution of receptor, which we did not observe experimentally. As experimental validation, we treated cells with blebbistatin (**Figure 5C**), an inhibitor of myosin II ATPases, and observed reduced peripheral OPN⁺ cell percentages (**Figure 5D**). These observations are in agreement with our previous experiments (**Kourouklis et al., 2016**), indicating that myosin-mediated contractility is necessary for peripheral biliary differentiation. TFM measurements obtained in parallel indicated loss of peripheral traction stresses in cells treated with blebbistatin (**Figure 5D**), in agreement with the known action of this inhibitor. These simulations demonstrate that a simple model of Notch *trans*-activation coupled with an external stress gradient is consistent with our experimental findings.

In order to ascertain which specific mechanotransduction pathways are involved in this process, we treated cells with inhibitors for ERK (FR180204) and ROCK (Y-27632) (**Figure 6A**). We observed that FR180204 reduced OPN⁺ cell percentages at the outer edge of the patterned domains (**Figure 6B**), which is in accordance with our previous studies suggesting involvement of ERK in biliary differentiation (**Kourouklis et al., 2016**). In contrast, inhibition of ROCK resulted in increased

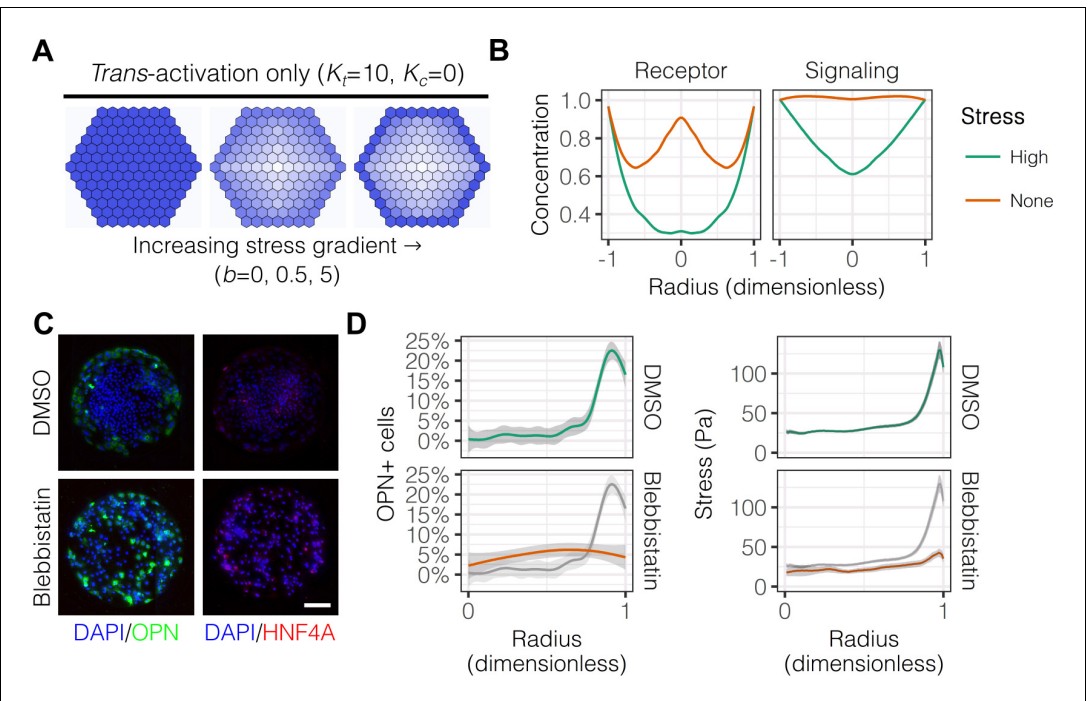

**Figure 5.** Peripheral differentiation is dependent on a gradient of actomyosin contractility. (**A**) Simulated effect of stress gradients of increasing steepness on Notch signaling activity via *trans*-activation. Darker shades of blue represent increased Notch signaling activity as measured by repressor levels. (**B**) Quantification of simulated Notch receptor and cleavage events generating repressor, a measure of Notch signaling activity, under conditions of no ($b = 0$) and high ($b = 5$) stress gradients. Concentration is in arbitrary units normalized to the periphery for each condition. (**C**) BMEL cells on 30 kPa substrates were presented with DLL4 and treated with vehicle control (DMSO) and inhibitor of myosin II ATPases (blebbistatin, 25 μM). Samples were immunolabeled for OPN and HNF4A. Scale bar is 150 μm. (**D**) Quantification of OPN[+] cell percentages and stress from TFM after treatment with DMSO and blebbistatin. Gray lines and associated ribbons represent the DMSO control replotted in additional panels to aid comparison. Mean ± 95% CI.

DOI: https://doi.org/10.7554/eLife.38536.018

peripheral OPN[+] cell percentages extending centrally (*Figure 6A and B*). Consistent with the respective functions of the proteins targeted by these inhibitors, TFM measurements indicated loss of peripheral traction stresses in cells treated with Y-27632 but not FR180204 (*Figure 6C*). Analysis of *Jag1* and *Notch2* mRNA expression in cells treated with FR180204 indicated that inhibition of ERK signaling results in direct reduction in expression of both ligand and receptor (*Figure 6D*). Furthermore, we observed that the Hippo pathway effector YAP exhibited increased expression at the periphery of arrayed patterns on both 30 kPa and 4 kPa substrates (*Figure 6E*), though the expression of YAP was not altered by the presence of Notch ligands in the arrayed domains (data not shown). This observation of peripheral YAP expression is especially interesting in light of recent findings regarding the demonstrated role of YAP in biliary fate (*Yimlamai et al., 2014*) and suggests a potential role for the Hippo pathway in progenitor fate segregation. Collectively, these data show that peripheral biliary differentiation is dependent on myosin-mediated cell contractility and ERK signaling and is decoupled from mechanical stress when ROCK is inhibited.

## Notch ligands *Jag1* and *Dll1* are necessary for fate segregation

Studies of Alagille syndrome, a genetic disorder which results in bile duct paucity, have shown that the Notch ligand *JAG1* is necessary for bile duct formation (*Li et al., 1997*; *Oda et al., 1997*). Our previous work has also shown that the Notch ligand *Dll1* can modulate differentiation toward both biliary and hepatocytic fates (*Kaylan et al., 2016*). We therefore hypothesized that the Notch ligands *Jag1* and *Dll1* are involved in the segregation of liver progenitor fates in arrayed patterns.

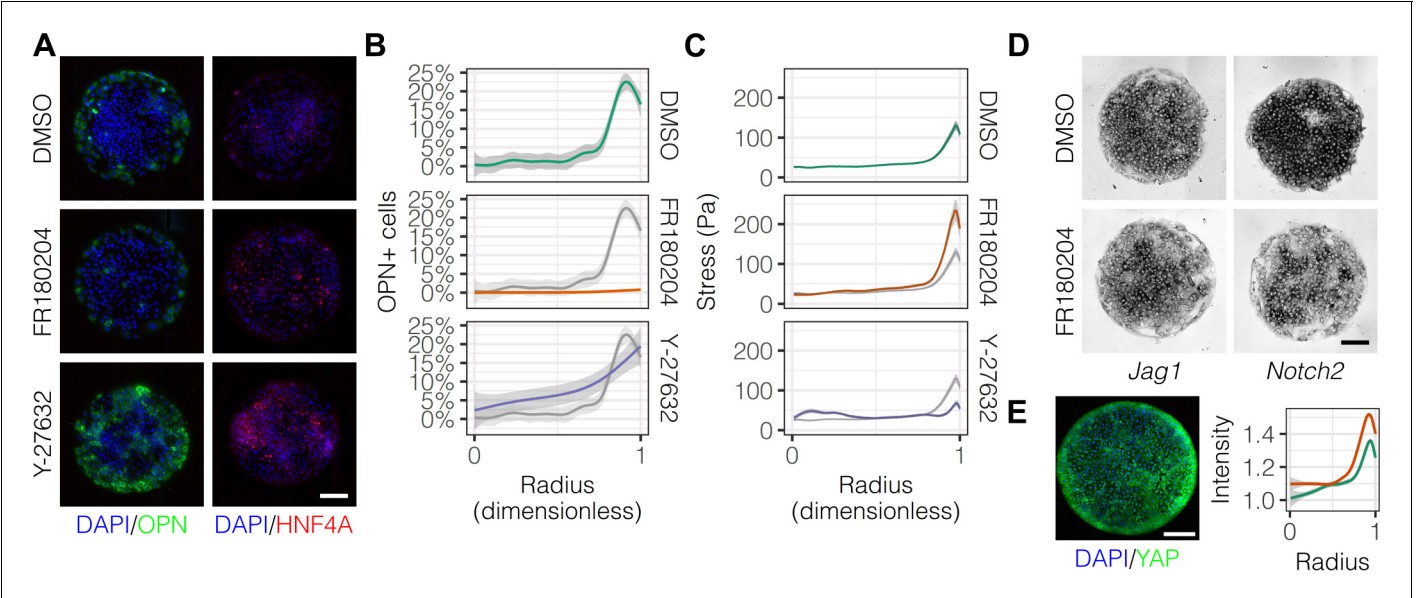

**Figure 6.** Mechanotransduction by ERK and ROCK modulate peripheral biliary fate. (**A**) BMEL cells on 30 kPa substrates were presented with DLL4 and treated with vehicle control (DMSO) and inhibitors of ERK signaling (FR180204, 10 μM) and ROCK (Y-27632, 10 μM). Samples were immunolabeled for OPN and HNF4A. (**B**) Quantification of OPN$^+$ cell percentages after treatment with DMSO, FR180204, or Y-27632. (**C**) Stress from TFM after treatment with DMSO, FR180204, or Y-27632. (**D**) RNA in situ hybridization for *Jag1* and *Notch2* of cells exogenously presented with DLL4 and treated with DMSO or FR180204. (**E**) Immunolabeling for YAP of BMEL cells presented with DLL4 on 30 kPa substrates. Quantification shows YAP intensity (a.u.) of cells presented with DLL4 on 30 kPa (green) and 4 kPa (orange) substrates by dimensionless radius. Mean ± 95% CI. (**A, D, E**) Scale bars are 150 μm. (**B, C**) Gray lines and associated ribbons represent the DMSO control replotted in additional panels to aid comparison. Mean ± 95% CI.

DOI: https://doi.org/10.7554/eLife.38536.019

The following source data is available for figure 6:

**Source data 1.** Summary table for OPN$^+$ percentage data in *Figure 5D* and *Figure 6B*.
DOI: https://doi.org/10.7554/eLife.38536.020

**Source data 2.** Summary table for TFM data in *Figure 4D*, *Figure 4—figure supplement 1*, *Figure 4—figure supplement 2*, *Figure 5D*, and *Figure 6C*.
DOI: https://doi.org/10.7554/eLife.38536.021

Using lentiviral shRNA vectors, we knocked down *Jag1* (shJag1) and *Dll1* (shDll1) in liver progenitors and cultured them on arrayed patterns (*Figure 7A and B*). We observed that shJag1 cells exhibited reduced OPN$^+$ cell counts at the periphery ($\beta = -16.3$, $P < 0.001$) while, in contrast, counts of peripheral shDll1 cells that were OPN$^+$ increased ($\beta = 20.1$, $P < 0.001$), observations confirmed by quantification (*Figure 7C*) and regression analysis (*Figure 7—figure supplement 1*). In agreement with the data for OPN, only shJag1 cells exhibited loss of peripheral SOX9 expression (*Figure 7D*). Interestingly, knockdown of both *Jag1* and *Dll1* resulted in decreased central HNF4A expression (*Figure 7B and D*). TFM of shJag1 and shDll1 cells showed no reduction in cell-generated traction stresses by ligand knockdown compared to control cells (data not shown). These data establish contrasting roles for *Jag1* and *Dll1* in biliary differentiation in which *Dll1* has the unanticipated function of antagonizing biliary fate and, further, suggest that the ligands are involved in hepatocytic differentiation of progenitor cells.

## Discussion

Here, we utilized microarrayed patterns of ECM co-presented with Notch ligands to provide a biochemically- and biophysically-defined microenvironment for liver progenitor differentiation. In these patterns, we observed spatially-localized, segregated differentiation of progenitors toward biliary fates peripherally and hepatocytic fates centrally. Other groups have made similar observations using both 2D and 3D engineered systems as part of studies investigating the differentiation of mesenchymal and induced pluripotent stem cells (*Ruiz and Chen, 2008*; *Kilian et al., 2010*; *Ma et al., 2015*; *Lee et al., 2015*). In these other cell types, pathways related to cell contractility (*e.g.*, RhoA,

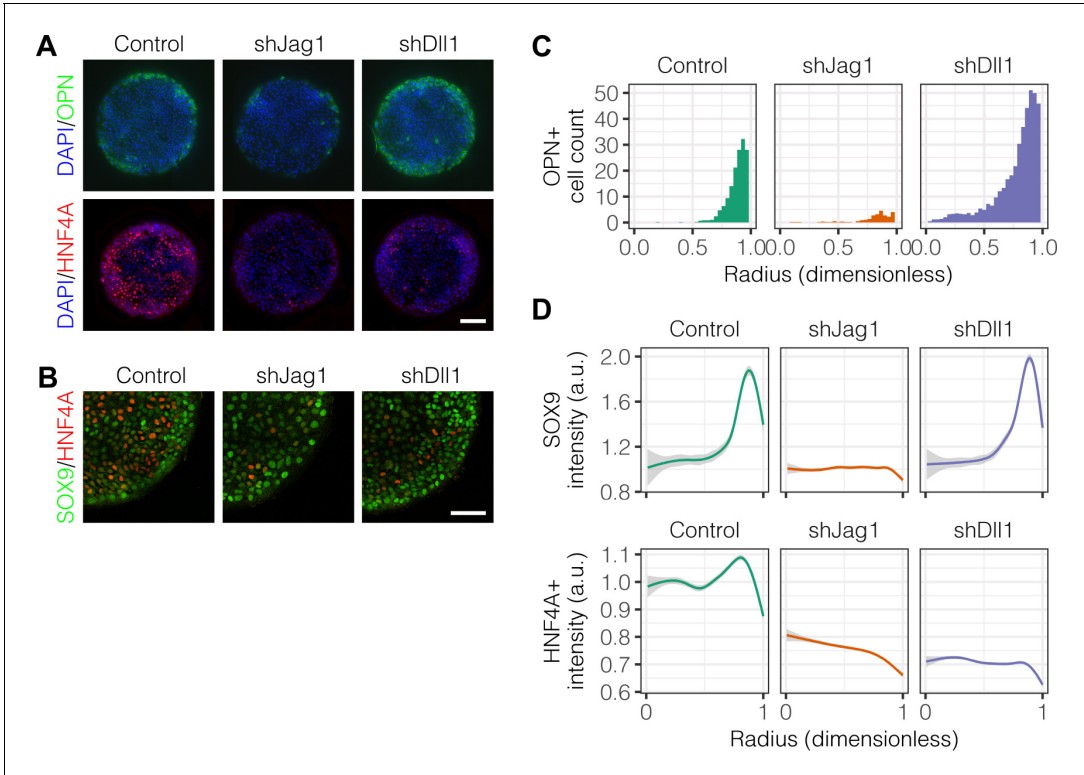

**Figure 7.** Notch ligands *Jag1* and *Dll1* are both required for segregation of hepatocytic fate centrally and biliary fate peripherally. (**A**) Immunolabeling for OPN and HNF4A of BMEL cells presented with DLL4 on 30 kPa substrates. Control cells were transduced with an shRNA vector coding for a non-mammalian target. shJag1 and shDll1 cells were transduced with shRNA vectors targeting *Jag1* and *Dll1*, respectively. Scale bar is 150 μm. (**B**) Confocal imaging of immunolabeled SOX9 and HNF4A in control, shJag1, and shDll1 cells presented with DLL4 on 30 kPa substrates. Scale bar is 75 μm. (**C**) Quantification of OPN$^+$ cell counts of control, shJag1, and shDll1 cells presented with DLL4 on 30 kPa substrates. (**D**) Quantification of SOX9 and HNF4A intensity of control, shJag1, and shDll1 cells presented with DLL4 on 30 kPa substrates. (**C, D**) Mean ± 95% CI.

DOI: https://doi.org/10.7554/eLife.38536.022

The following source data and figure supplement are available for figure 7:

**Source data 1.** Summary table for OPN data in *Figure 7C*.

DOI: https://doi.org/10.7554/eLife.38536.024

**Source data 2.** Summary table for SOX9 and HNF4A data in *Figure 7D*.

DOI: https://doi.org/10.7554/eLife.38536.025

**Figure supplement 1.** Regression analysis of OPN+ cell counts.

DOI: https://doi.org/10.7554/eLife.38536.023

ROCK, RAC1) and cell–cell adaptor proteins (*e.g.*, E-cadherin) were both implicated. We show in this work that cell contractility is a key inducer of biliary fate in liver progenitors and elaborate roles for cell–cell interactions and mechanotransduction pathway activity in addition to established regulation by Notch and TGFβ signaling.

We have previously examined the role of substrate stiffness in the context of TGFβ-induced biliary differentiation, finding that progenitor cells cultured on fibronectin are sensitive to stiffness whereas cells cultured on collagen IV differentiated independent of stiffness (*Kourouklis et al., 2016*). On collagen I patterns, we observed that high substrate stiffness ($E \sim 30\,\mathrm{kPa}$) increases peripheral biliary differentiation, Notch family member expression, and responsiveness to cell-extrinsic ligand presentation (*Figure 2*). In contrast, low substrate stiffness ($E \sim 4\,\mathrm{kPa}$) was more supportive of hepatocytic fate, particularly in the pattern center (*Figure 1* and *Figure 2*). These findings are consistent with other recent efforts toward delineating the impact of substrate stiffness on hepatocyte function, which have identified potential mechanisms of transcriptional and epigenetic repression of HNF4A in

hepatocytes experiencing increased cytoskeletal tension (*Desai et al., 2016*; *Cozzolino et al., 2016*).

By integrating TFM with the array platform, we were able to localize traction stresses and associated cell contractility to the pattern periphery, coincident with biliary differentiation (*Figure 4*). Paradoxically, treatment with inhibitors of actomyosin contractility (blebbistatin and Y-27632) resulted in divergent fate trajectories. Blebbistatin, a direct inhibitor of myosin ATPase, reduced both peripheral traction stress and downstream biliary differentiation as expected, whereas Y-27632, an inhibitor of myosin light chain phosphorylation by ROCK, increased peripheral differentiation and extension of differentiation centrally (*Figure 5* and *Figure 6*). It is possible this divergence is due to the antagonism of ROCK against RAC1-induced adherens-junction formation (*Wildenberg et al., 2006*), suggesting that increased cell–cell interactions in the context of reduced cytoskeletal tension is supportive of biliary fate. Further evidence for this hypothesis is our observation of uniform induction of biliary fate by DLL4 presentation in cells with adherens junctions inhibited by DECMA (*Figure 3*), results which raise the additional possibility that Notch ligand–receptor binding is dependent on adherens junction formation. Lowell *et al*. provide evidence of such a mechanism in human keratinocytes, observing mutual exclusion of E-cadherin and Delta ligand and further noting that ligand expression promotes cell–cell interactions independent of adherens junction formation (*Lowell et al., 2000*).

Interestingly, treatment with an inhibitor of TGFβ (SB-431542) reduced biliary differentiation and increased hepatocytic differentiation but failed to abolish peripheral expression of *Jag1* and *Notch2* (*Figure 3*). It is therefore not likely that TGFβ signaling is the single factor responsible for peripheral biliary fate and associated gradient formation, though it may act through autocrine or paracrine regulation to enable differentiation by other mechanisms. For instance, Zavadil *et al*. showed that TGFβ serves as a leading signal in the biphasic activation of HEY1 *via* interactions with SMAD3 and SMAD4 transcriptional regulators, whereas the lagging signal consisted of sustained HEY1-mediated activation of JAG1 signaling dependent on ERK (*Zavadil et al., 2004*). In the context of liver progenitor fate, this model would require only moderate amounts of autocrine TGFβ to activate the Notch transcriptional machinery leading to ligand expression and associated biliary differentiation. In support of this model, inhibition of ERK signaling with FR180204 reduced both biliary differentiation and peripheral expression of *Jag1* and *Notch2* (*Figure 6*). Last, our observation of peripherally-expressed cytoplasmic YAP (*Figure 6*) is intriguing in light of recent literature regarding the role of YAP as a mechanosensor (*Dupont et al., 2011*) and regulator of liver cell fate (*Yimlamai et al., 2014*; *Lee et al., 2016*) and might serve as a mechanistic effector downstream of peripherally-induced cytoskeletal tension in progenitor cells.

Our observations of peripherally-expressed *Jag1*, *Dll1*, and *Notch2* (*Figure 2*) are especially striking in light of the TFM data showing colocalization with peak traction stresses. Although we have demonstrated dependence of peripheral expression of ligand and receptor on substrate stiffness and ERK signaling, the exact mechanism linking traction stress to Notch ligand and receptor expression remains unidentified. Answering this question is crucial in order to define the role of cytoskeletal stress relative to Notch and TGFβ in biliary differentiation of liver progenitors. TFM of progenitor cells treated with GSI places generation of traction stresses prior to Notch-mediated biliary differentiation (*Figure 4—figure supplement 1*). In contrast, TFM of cells treated with SB-431542 provides evidence that TGFβ is upstream of traction stress (*Figure 4—figure supplement 2*), in accordance with the biphasic model described above in which TGFβ serves as an initial stimulus to Notch activity as well as potential feed-forward induction of cell contractility by TGFβ under conditions of mechanical stress (*Tomasek et al., 2002*). Recent descriptions of new modes of non-canonical Notch signaling provide other potential mechanisms linking cytoskeletal stress and Notch through ligand–intermediate filament interactions (*Antfolk et al., 2017*) or Notch transmembrane domain-mediated activation of RAC1 signaling (*Polacheck et al., 2017*).

To gain insight into how cell mechanical stress may influence the Notch pathway, we explored the utility of incorporating mechanical stress into a multicellular model of Notch pathway dynamics (*Figure 5*). The results of this integrated model demonstrate that the introduction of mechanical stress as a positive regulator of Notch receptor and Notch ligand expression is sufficient to generate a patterning response with enhanced peripheral Notch activation. Notably from the in situ hybridization experiments, the presence of the Notch ligand DLL4 in the arrayed domains appeared to enhance central expression of *Jag1* and *Notch2* mRNA on 30 kPa but not 4 kPa substrates (*Figure 2*).

This observation would suggest that DLL4 is acting to enhance Notch signaling centrally on 30 kPa. However, cells presented with DLL4 on 30 kPa substrates exhibited preferential biliary differentiation at the periphery with minimal biliary differentiation centrally, indicating that central expression of Notch pathway components may not be sufficient for biliary differentiation. Taken together, these findings suggest that the spatial distribution of mechanical stress signals may impact cell differentiation not only by influencing the expression of Notch pathway members but also through interactions with downstream Notch-mediated transcription or through cooperation with TGFβ and ERK, which is required for differentiation. Furthermore, future experiments incorporating additional quantitative measurements of spatial mRNA expression will be useful in identifying subtler patterns of Notch ligand and receptor expression.

Knockdown of cell-intrinsic *Jag1* and *Dll1* further revealed distinct roles in both biliary and hepatocytic differentiation of progenitor cells (*Figure 7*). The reduction of central HNF4A with knockdown of either ligand is particularly interesting and suggests a role for cell–cell interactions with ligand-presenting cells in hepatocytic differentiation. The loss of biliary differentiation with *Jag1* knockdown is consistent with the known role of *Jag1* expressed in the mesenchyme of the portal vein (*Hofmann et al., 2010*). The unanticipated increase in OPN$^+$ cells as a consequence of *Dll1* knockdown, however, has fewer precedents and suggests an cell-intrinsic inhibitory role in contrast with that of *Jag1*. Although we used multiple Notch ligands (DLL1, DLL4, JAG1) in arrays, we have largely focused on presentation of DLL4 to progenitor cells due to its consistent activation of progenitor cells. The differential cell-extrinsic activity of the ligands might be explained in part by the known preferential affinity of ligands for specific receptors (*Yamamoto et al., 2012*; *Andrawes et al., 2013*) as well as recent evidence showing that DLL4 binds Notch receptors with greater affinity and requires less mechanical tension to activate signaling (*Luca et al., 2017*). It may also be a consequence of ligand presentation in the array format, which is known to be a function of molecular weight and charge (*Flaim et al., 2005*; *Reticker-Flynn et al., 2012*).

Despite the previously established role of substrate stiffness in hepatocellular differentiation, one of the unexpected observations of these studies was the significant cooperative effect that substrate stiffness exhibited with multicellular geometry. Although substrate stiffness did not substantially influence mechanical stress profiles as measured by TFM (*Figure 4*), substrate stiffness altered the baseline levels of hepatocyte and biliary markers, with stiffer substrates promoting biliary differentiation and reducing hepatocyte differentiation (*Figure 2*). Overall, this observation highlights the importance of considering tissue stiffness as a potential variable within current and future studies examining other regulatory signals, such as Notch. In addition, future studies could examine a broader range of geometries, including non-circular. Our analysis of different pattern sizes suggested that peripheral differentiation was independent of diameter (*Figure 1—figure supplement 3*). As a result, with decreasing diameter, a greater fraction of the cells are at the periphery and exhibit biliary differentiation. Subsequent studies could be aimed at further reducing pattern size or even patterning single cells to determine if there is a size that balances mechanical stress and other intercellular signals for achieving optimal biliary differentiation.

Although the array patterns we used represent a relatively simple 2D geometry, we anticipate that the mechanisms regulating progenitor differentiation investigated here will serve as a foundation for future efforts employing 3D culture models while also helping to identify candidates for future manipulation in vivo. Interestingly, during liver development, biliary differentiation is initiated as a ductal plate consisting of a layer of differentiating progenitor cells that encircle the portal vein (*Ober and Lemaigre, 2018*). Based on our findings related to the spatial patterning of progenitor differentiation, it is reasonable to hypothesize that the structure of the portal vein may play a role in defining the geometric and mechanical cues presented to the nascent biliary cells. In these studies, we utilized BMEL cells, which are untransformed and have been demonstrated to exhibit bipotential differentiation both in vitro and in vivo (*Strick-Marchand and Weiss, 2002*; *Strick-Marchand et al., 2004*). Accordingly, they represent a robust model cell type for controlled in vitro studies investigating microenvironmental regulation of progenitor fate specification. Building on our findings presented here, the cellular microarray approach could be adapted for investigating the differentiation of immortalized human bipotential cell lines and, ultimately, primary or stem cell-derived human liver progenitors.

Finally, the mechanoresponsiveness of liver progenitors has crucial implications not only for development but also disease. Cholangiocytic cells derived from transitional progenitors have been

implicated in the pathogenesis of cholangiopathies, cholangiocarcinomas, and related disorders through compensatory ductular reactions (*Gouw et al., 2011*) and are further thought to play a role in regenerating the liver by transdifferentiation (*Boulter et al., 2012*; *He et al., 2014*; *Raven et al., 2017*). The mechanisms we describe here may contribute to early sensing of and differentiation responses to the stiff, fibrotic microenvironments in both ductular reactions and regeneration, contributing to the biliary fates observed in these contexts.

## Materials and methods

### Cell culture

We utilized BMEL 9A1 cells between passages 30 and 36. These cells were cultured as previously described (*Strick-Marchand and Weiss, 2002*). Briefly, cells were seeded on tissue culture plastic coated with collagen I (0.5 mg/ml) and subsequently cultured under controlled environmental conditions (37°C and 5% $CO_2$). Treatment with trypsin-EDTA (0.25% v/v) for $\leq$10 min was used to detach cells for subculturing. Basal media for expansion consisted of RPMI 1640 with fetal bovine serum (10% v/v, FBS), penicillin/streptomycin (1% v/v, P/S), L-glutamine (1% v/v), human recombinant insulin (10 µg/ml, Life Technologies, 12585–014), IGF-2 (30 ng/ml, PeproTech, 100–12), and EGF (50 ng/ml, PeproTech, AF-100–15). Differentiation media consisted of Advanced RPMI 1640 (Life Technologies, 12633–012) with FBS (2% v/v), P/S (0.5% v/v), L-glutamine (1% v/v), and minimum non-essential amino acids (1% v/v, Life Technologies, 11140–050). BMEL cells tested negative for *Mycoplasma* spp. using the MycoProbe Mycoplasma Detection Kit (R&D Systems, #CUL001B). We confirmed expression of liver-specific genes and proteins in bulk cultures using PCR, immunocytochemistry, and western blot as previously described (*Strick-Marchand and Weiss, 2002*; *Kaylan et al., 2016*; *Kourouklis et al., 2016*). Additionally, bipotential differentiation capacity of BMEL cells was confirmed using bulk cultures within standard tissue culture plates with or without treatment with TGFβ1 (*Kaylan et al., 2016*; *Kourouklis et al., 2016*). During microarray-based differentiation experiments, cells were seeded on arrays at 1E6 cells/slide (immunocytochemistry) and 500E3 cells/dish (TFM). Cells were allowed to adhere to arrays for 2 hr before addition before 2× washes with differentiation media and subsequent addition of experiment-specific treatments. All growth factors and drugs used in these experiments were prepared and reconstituted according to the instructions of the manufacturers; see *Table 1*. The control, shJag1, and shDll1 cells were generated by lentiviral transduction with shRNA constructs targeting a non-mammalian sequence, *Jag1*, and *Dll1*, respectively, the details and validation of which we have described elsewhere (*Kaylan et al., 2016*).

### Preparation of polyacrylamide hydrogels

Polyacrylamide (PA) hydrogels were prepared following previous protocols (*Aratyn-Schaus et al., 2010*; *Tse and Engler, 2010*; *Wen et al., 2014*). Briefly, 25×75 mm glass microscope slides were washed with 0.25% v/v Triton X-100 in $dH_2O$ and placed on an orbital shaker for 30 min. After rinsing with $dH_2O$, slides were immersed in acetone and placed on the shaker for 30 min. The acetone wash was followed by immersion in methanol and another 30 min on the shaker. The slides were then washed with 0.2 N NaOH for 1 hr, rinsed with $dH_2O$, air-dried, and placed on a hot plate at

**Table 1** List of growth factors and drugs.

| Factor or drug | Stock | Target | Manufacturer | Catalog # |
| --- | --- | --- | --- | --- |
| (–)-Blebbistatin | 1 mg/ml | 25 µM | Cayman Chemical | 13013 |
| DECMA | 1 mg/ml | 10 µg/ml | Fisher Scientific | 50-245-625 |
| FR180204 | 10 mg/ml | 10 µM | Sigma-Aldrich | SML0320 |
| L-685,458 (GSI) | 1 mM | 5 µM | Tocris | 2627 |
| SB-431542 | 10 mM | 10 µM | Sigma-Aldrich | S4317 |
| TGFβ1 | 5 µg/ml | 1.5 ng/ml | R&D Systems | 240-B-002 |
| Y-27632 | 5 mg/ml | 10 µM | Enzo Life Sciences | 270–333-M005 |

DOI: https://doi.org/10.7554/eLife.38536.030

110°C until dry. For silanization, the cleaned slides were immersed in 2% v/v 3-(trimethoxysilyl)propyl methacrylate in ethanol and placed on the shaker for 30 min. The silanized slides were washed with ethanol on the shaker for 5 min, air-dried, and again placed on the hot plate at 110°C until dry. For fabrication of hydrogels with specific elastic moduli, two prepolymer solutions with different acrylamide/bis-acrylamide percentage (w/v) ratios were prepared to achieve elastic moduli of 4 kPa (4% acrylamide, 0.4% bis-acrylamide) and 30 kPa (8% acrylamide, 0.55% bis-acrylamide) with similar porosity (Wen et al., 2014). Each of these prepolymer solutions were mixed with Irgacure 2959 (BASF, Corp.) solution (20% w/v in methanol) at a final volumetric ratio of 9:1 (prepolymer:Irgacure). This working solution was then deposited onto slides (100 µl/slide) and covered with 22×60 mm cover glasses. The sandwiched working solution was transferred to a UV oven and exposed to 365 nm UV A for 10 min (240E3 µJ). After removing the cover glasses, the slides were immersed in dH$_2$O at room temperature for 3 d in order to remove excess reagents from the hydrogel substrates. Before microarray fabrication, hydrogel substrates were thoroughly dehydrated on a hot plate for ≥15 min at 50°C.

## Microarray fabrication

Microarrays were fabricated as described previously (Flaim et al., 2005; Brafman et al., 2012; Kaylan et al., 2016). Biomolecules for arraying were diluted in 2× growth factor buffer (38% v/v glycerol in 1× phosphate-buffered saline [PBS], 10.55 mg/ml sodium acetate, 3.72 mg/ml EDTA, 10 mg/ml CHAPS) and loaded in a 384-well V-bottom microplate. Collagen I (rat tail, EMD Millipore, 08–115) was prepared at a final concent µg/ml. Fc-recombinant Notch ligand solutions were prepared at a final concentration of 104 µg/ml and included: Fc-JAG1 (R&D Systems, 599-JG-100), Fc-DLL1 (R&D Systems, 5026-DL-050), and Fc-DLL4 (Adipogen, AG-40A-0145-C050). All Notch ligand conditions were pre-conjugated with Protein A/G (Life Technologies, 21186) at a minimum 1:6 molar ratio (A/G:ligand) before arraying. Human IgG (104 µg/ml final, R&D Systems, 1–001-A) was arrayed as a control in experiments involving Notch ligands. A robotic benchtop microarrayer (OmniGrid Micro, Digilab) loaded with SMPC Stealth microarray pins (ArrayIt) was used to transfer biomolecules from source plate to polyacrylamide hydrogel substrate, producing ~600 µm diameter arrayed domains. For other pattern sizes, we used Xtend pins (LabNEXT) at 200 µm and 700 µm diameter. Fabricated arrays were stored at room temperature and 65% RH overnight and left to dry under ambient conditions in the dark. Prior to cell culture, the arrays were sterilized with 30 min UVC while immersed in 1× PBS supplemented with 1% (v/v) P/S, after which cells were seeded on arrays as described above.

## Image processing and analysis of microarrays

Images of entire arrays were converted to individual 8-bit TIFF files per channel (i.e., red, green, blue, and gray) by Fiji (ImageJ version 1.51n) (Schneider et al., 2012; Schindelin et al., 2012). Image size was reduced to ~50 megapixels/channel by binning to reduce memory requirements during computational analysis. The IdentifyPrimaryObjects and IdentifySecondaryObjects modules of CellProfiler (version 2.2.0) (Kamentsky et al., 2011) were used to identify nuclei for cell counts and regions marked by fluorescence. The MeasureObjectIntensity module was used to quantify single-cell intensity. The location of arrayed conditions within each image was automatically determined relative to manually-located dextran-rhodamine markers. The centroid of each island was calculated and used to assign a radial distance to each cell for analyses of spatial localization within arrayed patterns.

## Immunocytochemistry

Samples were fixed in paraformaldehyde (4% w/v in 1× PBS) for 15 min. Samples intended for labeling of secreted proteins (namely ALB and OPN) were treated with brefeldin A (10 µg/ml, R&D Systems, 1231/5) for 2 hr prior to fixation. Fixed samples were permeabilized with Triton X-100 (0.25% v/v in 1× PBS) for 10 min and incubated in blocking buffer (5% v/v donkey serum and 0.1% v/v Triton X-100 in 1× PBS) for 1 hr at room temperature. We incubated samples for 1 hr at room temperature or overnight at 4°C with one or two of the primary antibodies listed in Table 2 diluted in blocking buffer. The next day, we incubated samples for 1 hr at room temperature with one or two of the following secondary antibodies diluted in blocking buffer: DyLight 488-conjugated donkey anti-rabbit

**Table 2** List of primary antibodies.

| Antibody target | Dilution | Manufacturer | Catalog # |
|---|---|---|---|
| ALB | 1/100 | Bethyl | A90-134A |
| CK19 | 1/200 | Abcam | ab52625 |
| Digoxigenin | 1/500 | Roche | 11 093 274 910 |
| HNF4A | 1/200 | Abcam | ab41898 |
| OPN (SPP1) | 1/50 | R&D Systems | AF808 |
| SOX9 | 1/200 | EMD Millipore | AB5535 |
| YAP1 | 1/50 | ProteinTech | 13584–1-AP |

DOI: https://doi.org/10.7554/eLife.38536.031

IgG (1/50 from stock, Abcam, ab96919), DyLight 550-conjugated donkey anti-mouse IgG (1/50 from stock, Abcam, ab98767), and DyLight 488-conjugated donkey anti-goat IgG (1/50 from stock, Abcam, ab96935). Samples were mounted in Fluoromount G with DAPI (Southern Biotech, 0100–20) and imaged no earlier than the day after mounting using an Axiovert 200M microscope (Carl Zeiss, Inc.) and associated Zen Pro software. In order to capture entire arrays as one image for later analyses, we utilized the tiling feature of Zen Pro.

## mRNA in situ hybridization

We performed in situ hybridization as previously-described (*Biehl and Raetzman, 2015*; *Aujla et al., 2015*). Samples were fixed in paraformaldehyde (4% w/v in 1× PBS) for 10 min, permeabilized with 0.3% Triton X-100 in 1× PBS for 15 min, and digested with Proteinase K (0.1 µg/ml) for 15 min at 37°C. Afterwards, samples were acetylated, pre-hybridized, and incubated in hybridization solution with linearized, digoxigenin-labeled probes for *Jag1*, *Dll1*, or *Notch2* at 55°C. Prior to initiation of hybridization, probes were denatured for 3 min at 95°C. After overnight incubation, samples were washed in 50% 0.5× formamide solution and 0.5× sodium citrate and subsequently blocked (10% heat-inactivated sheep serum, 2% bovine serum albumin and 0.1% Triton X-100 in tris-buffered saline). Following blocking, slides were incubated with anti-digoxigenin antibody (see *Table 2*) diluted in blocking buffer for 1 hr. Next, samples were washed with tris-buffered saline of increasing alkalinity (pH = 7.5, 9.5) and incubated overnight in NBT/BCIP developing solution (Roche, 11 681 451 001). Samples were subsequently fixed with paraformaldehyde (4% w/v in 1× PBS for 10 min), mounted in Fluoromount G with DAPI (Southern Biotech, 0100–20), and imaged similarly to the immunofluorescently-labeled samples described above.

## Traction force microscopy

For TFM experiments, we adjusted our protocol in order to fabricate the PA hydrogels in glass-bottom 35 mm Petri dishes (Cell E&G, GBD00002-200) rather than on 25×75 mm microscope slides. This enabled us to perform TFM on live cells at 37°C and 5% CO$_2$. To measure the cell-generated forces, we added 1 µm far-red fluorescent beads (0.2% v/v, Life Technologies, F-8816) to the working solution (*Wang and Lin, 2007*; *Wang et al., 2002*) and fabricated hydrogels with embedded beads by exposure to 365 nm UV A for 10 min. We subsequently completed the hydrogel and array fabrication protocols as described above and seeded cells on the arrays. After completion of experiment-specific treatments, the arrays were transferred to an incubated (37°C and 5% CO$_2$) Axiovert 200M microscope (Carl Zeiss, Inc.). The microscope was used to capture phase contrast and far-red fluorescent micrographs to record cellular position and morphology along with bead displacement before and after cell dissociation with sodium dodecyl sulfate (1% v/v in 1× PBS). For analysis, we calculated the traction fields from the displacements using standard methods (*Butler et al., 2002*; *Wang et al., 2002*) which we have adapted for analysis of cell microarrays elsewhere (*Kaylan et al., 2017*). We next analyzed the captured images in MATLAB software (MathWorks, Inc) using Bio-Formats (*Linkert et al., 2010*) in conjunction with a set of custom scripts (see *Source code 1–7*). Specifically, the border of each island was identified, allowing calculation of a best fit ellipse and centroid. A previously-described digital image correlation program was used to calculate the displacement field between the contracted and relaxed state (*Bar-Kochba et al., 2015*).

## Finite element analysis

Cell island contraction was simulated using COMSOL Multiphysics software (COMSOL Inc., Burlington, MA) as already described (*Nelson et al., 2005*) using previously-determined parameter values (*Sato et al., 1990*; *Folkman and Moscona, 1978*). Briefly, the model was comprised of an active layer bound to a passive substrate with fixed lower boundary. The cell island (20 μm height, 600 μm diameter) was modeled as an isotropic linearly-elastic material with Young's modulus of 1.5 kPa, Poisson's ratio of 0.48, thermal conductivity of 10 $Wm^{-1}K^{-1}$, and coefficient of expansion of 0.05 $K^{-1}$. The substrate was modeled as an isotropic linearly-elastic material with Young's modulus of 30 kPa and Poisson's ratio of 0.48. Contraction was induced in the model by reducing the temperature by 5 K (see *Source code 8*).

## Notch simulations

Our computational model for Notch signaling is based on that of the groups of Elowitz and Sprinzak (*Sprinzak et al., 2010*; *Formosa-Jordan and Sprinzak, 2014*), extending their approach to include the effect of an external gradient of a morphogen which regulates expression of Notch ligand and receptor. The model outputs a hexagonal lattice with fixed (rather than periodic) boundaries containing individual cells with their respective Notch, Delta, and repressor concentrations as determined by the following equations:

$$\frac{dN_i}{d\tau} = \alpha_n - K_t N_i \langle D_i \rangle - K_c N_i D_i - \gamma_n N_i + \sigma(b) \tag{1}$$

$$\frac{dD_i}{d\tau} = \frac{\alpha_d}{1 + \left(\frac{R_i}{\theta_r}\right)^h} - K_t D_i \langle N_i \rangle - K_c N_i D_i - \gamma_d D_i + \sigma(b) \tag{2}$$

$$\frac{dR_i}{d\tau} = \frac{\alpha_r \left(\frac{K_t N_i \langle D_i \rangle}{\gamma_{nd}}\right)^m}{\theta_{nd}^m + \left(\frac{K_t N_i \langle D_i \rangle}{\gamma_{nd}}\right)^m} - \gamma_R R_i \tag{3}$$

Where $N$ is Notch receptor concentration, $D$ is Delta ligand concentration, $R$ is repressor concentration, $\sigma$ is the stress gradient function, $K_c$ is the constant representing strength of cis-interactions, $K_t$ is the constant representing strength of *trans*-interactions, $b$ is the base constant for steepness of the stress gradient, $\alpha$ is the maximal production rate, $\gamma$ is the maximal degradation rate, $h$ is the cooperativity of Delta inhibition, $m$ is the cooperativity of repressor activation, and $\theta$ is the Hill coefficient. Subscript $i$ indicates index within the hexagonal lattice while angle brackets denote ensemble value of neighbors of cell $i$. These equations were evaluated with and without the stress function ($\sigma$) under various strengths of *cis*- and *trans*-interactions. The model defines $\sigma$ to be a linear function of the radius of the island, thereby increasing expression of Notch ligand and receptor with radius in accordance with our mRNA in situ hybridization data (see *Source code 9–11*).

## Statistical testing

Array experiments consisted of at least three biological replicates with 18 total islands per combination of arrayed condition, treatment, cell type, and readout. Counts of cells positive for immunolabels are plotted as mean values representative of an individual island. Line plots of both percentages of positive cells and mechanical stress were calculated using local polynomial regression fitting and are shown with 95% CI ribbons in gray to allow for direct statistical comparisons, that is $P < 0.05$ if the 95% CI ribbons for two conditions do not overlap. The percentage of cells positive for an immunolabel (namely ALB and OPN) was calculated relative to cell counts in each of 30 radial bins across every island. Multiple regression analyses were performed in R using the base lm() function (R Core Team, 2017, R Foundation for Statistical Computing) and are presented as coefficient estimates (β) and associated 95% CI. All β coefficients in regressions represent mean changes in cell counts, for which positive β represents increased cell counts and negative β represents decreased cell counts. For each regression model, we confirmed homoscedasticity, normal distribution of residuals, and the absence of leveraged outliers using residual-fit, Q-Q, and scale-location plots. For select comparisons in the text, Welch's two-sample t-test was performed in R using the base t.test

function. For all hypothesis testing, $P < 0.05$ was considered significant and $P$-values below $P = 0.001$ are denoted as $P < 0.001$.

## Acknowledgements

The authors gratefully acknowledge Hélène Strick-Marchand and Mary C Weiss (Institut Pasteur) for providing BMEL cells. We would also like to thank Dianwen Zhang (Imaging Technology Group, Beckman Institute) for assistance with microarray imaging and Karen Weis (Molecular Physiology, University of Illinois at Urbana–Champaign) for advice and materials.

## Additional information

### Funding

| Funder | Grant reference number | Author |
|---|---|---|
| National Institute of Biomedical Imaging and Bioengineering | 5R03EB022254-02 | Gregory H Underhill |
| National Science Foundation | 1636175 | Gregory H Underhill |
| National Institute of Biomedical Imaging and Bioengineering | T32EB019944 | Ian C Berg |

The funders had no role in study design, data collection and interpretation, or the decision to submit the work for publication. The content is solely the responsibility of the authors and does not necessarily represent the official views of the National Institutes of Health.

### Author contributions

Kerim B Kaylan, Ian C Berg, Data curation, Software, Formal analysis, Validation, Investigation, Visualization, Methodology, Writing—original draft, Writing—review and editing; Matthew J Biehl, Resources, Validation, Investigation, Methodology, Writing—original draft, Writing—review and editing; Aidan Brougham-Cook, Data curation, Formal analysis, Investigation, Writing—review and editing; Ishita Jain, Formal analysis, Investigation, Writing—review and editing; Sameed M Jamil, Conceptualization, Data curation, Software, Formal analysis, Visualization, Methodology, Writing—review and editing; Lauren H Sargeant, Nicholas J Cornell, Investigation, Writing—review and editing; Lori T Raetzman, Conceptualization, Resources, Supervision, Project administration, Writing—review and editing; Gregory H Underhill, Conceptualization, Resources, Supervision, Funding acquisition, Validation, Methodology, Writing—original draft, Project administration, Writing—review and editing

### Author ORCIDs

Kerim B Kaylan (iD) http://orcid.org/0000-0001-7147-0614
Gregory H Underhill (iD) http://orcid.org/0000-0003-1002-5335

### Decision letter and Author response

Decision letter https://doi.org/10.7554/eLife.38536.045
Author response https://doi.org/10.7554/eLife.38536.046

## Additional files

### Supplementary files

• Source code 1. MATLAB function to process TFM images.
DOI: https://doi.org/10.7554/eLife.38536.032

• Source code 2. MATLAB function to analyze and plot TFM data.
DOI: https://doi.org/10.7554/eLife.38536.033

• Source code 3. MATLAB function to retrieve plane data.
DOI: https://doi.org/10.7554/eLife.38536.034

• Source code 4. MATLAB function to retrieve the reader for an image.
DOI: https://doi.org/10.7554/eLife.38536.035

• Source code 5. MATLAB function to draw boundaries around cells automatically.
DOI: https://doi.org/10.7554/eLife.38536.036

• Source code 6. MATLAB function to find the best fit of an ellipse for a given set of points.
DOI: https://doi.org/10.7554/eLife.38536.037

• Source code 7. MATLAB function to rotate and center cell boundaries for averaging.
DOI: https://doi.org/10.7554/eLife.38536.038

• Source code 8. COMSOL FEM simulation of cells on 30 kPa and 4 kPa substrates.
DOI: https://doi.org/10.7554/eLife.38536.039

• Source code 9. MATLAB Notch simulation for no stress (b=0).
DOI: https://doi.org/10.7554/eLife.38536.040

• Source code 10. MATLAB Notch simulation for intermediate stress (b=0.5).
DOI: https://doi.org/10.7554/eLife.38536.041

• Source code 11. MATLAB Notch simulation for high stress (b=5).
DOI: https://doi.org/10.7554/eLife.38536.042

• Transparent reporting form
DOI: https://doi.org/10.7554/eLife.38536.043

### Data availability

Source data tables (9 total) for the immunofluorescence and TFM array experiments are associated with the relevant figures. Source code files (11 total) have been included for the TFM analysis (Figure 4-6), FEM simulations (Figure 4), and Notch simulations (Figure 5). A detailed protocol for our array analysis technique together with source code has been made available elsewhere, see Kaylan et al. (J Vis Exp, 2017, e55362, http://dx.doi.org/10.3791/55362).

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
