## [Decision Letter]

Thank you for submitting your article "Spatial patterning of liver progenitor cell differentiation mediated by cellular contractility and Notch signaling" for consideration by *eLife*. Your article has been reviewed by two peer reviewers, and the evaluation has been overseen by a Reviewing Editor and Didier Stainier as the Senior Editor. The reviewers have opted to remain anonymous.

The reviewers have discussed the reviews with one another and the Reviewing Editor has drafted this decision to help you prepare a revised submission.

Summary:

The paper describes a compelling intersection between chemical and mechanical cues (Notch and TGFB signaling; cytoskeletal tension) during biliary specification, using an elegant interdisciplinary approach (computational tools, small molecule perturbation, biophysical measurements). To experimentally probe the role of mechanotransduction pathways, inhibitors of ERK and ROCK were used, as well as imaging of YAP localization. Notch ligands Jag1 and Dll1 were knocked down, which unexpectedly resulted in decreased hepatocyte differentiation. Based on these results, the authors assert that the Notch, TGFb and mechanical signals are involved in biliary differentiation.

Essential revisions:

1) Additional discussion is suggested to more directly establish the biological relevance of the experimental system (2D cell patterns vs. the complex 3D context of the developing liver). In line with this, the authors should comment on how predictably the gradients of soluble factors and the substrate stiffness affect the cell differentiation outcomes in their model, and how representative are the BMEL cells for primary human cells. Also, the role of the computational model is not clear, as it does not result in mechanistic insights, and its accuracy is uncertain.

2) Data were obtained for circular micropatterns 600 µm in diameter. Rationale for choosing this diameter, an analysis of the effects of pattern size on cell differentiation and a prediction of the maximum pattern size for pure cholangiocyte differentiation would be instructive.

3) Given the breadth of previous publications of the authors, additional commentary and analysis of the cell plating density and motility would help distinguish the current work. In particular, it should be explained what is the role of cell density on differentiation segmentation, and if there is a critical density for cell differentiation on the edges.

4) Some clarifications of the experimental data are suggested.

- For Figure 1D-F, the authors should clarify the ligand they used, consider moving the regression analysis from the supplement into the main figure (to replace the CK19 data), and explain why Alb^+^ cells were more prominent at R~0.5 rather than R=0.

- In Figure 2E, the authors show that Notch ligands are distributed at the periphery and this effect is disrupted by Dll4 presentation on stiff gels but not soft gels. It is not clear how the disruption of this distribution would enhance periphery differentiation.

- In Figure 3C, Dll1 should be included. Additionally, the statement "SB-431542 treatment reduced central induction of both Jag1 and Notch 2 by presentation with DLL4" should be explained.

5) To validate the claim that YAP plays a role in progenitor segmentation, the authors should try to quantify cytoplasmic YAP localization on soft (~ 4 kPa) substrates.

---

## [Author Response]

1) Additional discussion is suggested to more directly establish the biological relevance of the experimental system (2D cell patterns vs. the complex 3D context of the developing liver). In line with this, the authors should comment on how predictably the gradients of soluble factors and the substrate stiffness affect the cell differentiation outcomes in their model, and how representative are the BMEL cells for primary human cells. Also, the role of the computational model is not clear, as it does not result in mechanistic insights, and its accuracy is uncertain.

We thank the reviewers for these helpful critiques. Throughout the text, we have expanded the description of our rationale and justification for using the BMEL cells and arrayed 2D geometries, as well as a further discussion of how our results compare to initial predictions and previously established mechanisms. In addition, we have expanded our discussion of the included computational model, specifically highlighting the insights that this model, and models like it, could provide.

2) Data were obtained for circular micropatterns 600 µm in diameter. Rationale for choosing this diameter, an analysis of the effects of pattern size on cell differentiation and a prediction of the maximum pattern size for pure cholangiocyte differentiation would be instructive.

We have included a set of studies that specifically examine the effect of pattern diameter (new Figure 1—figure supplement 3). These findings demonstrated that OPN expression, indicative of biliary differentiation, remained confined to the periphery independent of pattern size. In addition to this new supplemental figure, an expanded discussion of these results and the potential influence of pattern diameter is included in the text.

3) Given the breadth of previous publications of the authors, additional commentary and analysis of the cell plating density and motility would help distinguish the current work. In particular, it should be explained what is the role of cell density on differentiation segmentation, and if there is a critical density for cell differentiation on the edges.

We agree that a further discussion of cell density and motility was warranted. We have added a new supplementary figure (Figure 1—figure supplement 2), which addresses these aspects. Specifically, we have included analysis of OPN and CK19 expression at an early time point (t = 24 h) that demonstrates that peripheral expression of these markers occurs very early following differentiation induction, suggesting that it is unlikely to be dependent on cell motility. Further, we quantified cell density across the patterned cellular islands. This quantification of cell density indicated uniform density with radius, ruling out condensation as the main mechanism of differentiation in this context.

4) Some clarifications of the experimental data are suggested.- For Figure 1D-F, the authors should clarify the ligand they used, consider moving the regression analysis from the supplement into the main figure (to replace the CK19 data), and explain why Alb^+^ cells were more prominent at R~0.5 rather than R=0.

We have clarified the legends for Figure 1D as well as what was previously Figure 1E–F (now Figure 1—figure supplement 1) by stating the ligand used in each subfigure. Specifically, we presented DLL4 in Figure 1D and IgG for the CK19 data now shown in Figure 1—figure supplement 1. We agree with the reviewers regarding the placement of the regression analysis of the OPN+ and ALB^+^ cell count data in Figure 1B–C. This regression analysis has therefore been moved to Figure 1E–F.

As for why ALB^+^ cells peak at R~0.5, this is a consequence of a reduction in ALB+ cells at the periphery with ligand presentation. In contrast, cells presented with IgG exhibit a monotonic distribution without reduction at the periphery (see the first panel on the left in Figure 1C). Although there is a visible reduction in ALB+ cells at the periphery in fluorescence images (see ALB row in Figure 1A), our linear regression analysis indicates that this reduction in peripheral ALB+ cells is only marginally significant (Figure 1F). We therefore chose not to discuss this specific result in detail.

- In Figure 2E, the authors show that Notch ligands are distributed at the periphery and this effect is disrupted by Dll4 presentation on stiff gels but not soft gels. It is not clear how the disruption of this distribution would enhance periphery differentiation.

The reviewers are correct that the distribution of Notch ligands is altered by the presence of DLL4 on the 30 kPa substrates. This is due to a central induction of Notch ligands on the 30 kPa substrates. However, on 4 kPa substrates, DLL4 does not similarly induce this increase of Notch ligand expression in the central region. Collectively, these observations imply that increasing substrate stiffness can induce further responsiveness to ligand presentation as measured by *Jag1* and *Notch2*. We believe this finding provides further support for our hypothesis that a biomechanical signal, such as substrate stiffness, can modulate Notch signaling in the context of liver progenitor differentiation. In regard to the resultant effect on peripheral differentiation, we hypothesize that the spatial distribution of mechanical stress signals may impact cell differentiation by not only influencing expression of Notch ligands/receptors but also by interacting with downstream Notch-mediated transcription or other cooperative pathways such as TGFβ and ERK signaling. Accordingly, despite an elevated level of Notch ligands/receptors centrally in islands on 30 kPa presented with DLL4, cell mechanical stress is still concentrated at the periphery of the islands resulting in peripheral progenitor biliary differentiation. We have added further clarification and discussion of these points to the Results and Discussion sections.

- In Figure 3C, Dll1 should be included. Additionally, the statement "SB-431542 treatment reduced central induction of both Jag1 and Notch 2 by presentation with DLL4" should be explained.

We appreciate the reviewers noting the lack of clarity in this statement and have amended it to read: “[…] SB-431542 treatment reduced expression of both *Jag1* and *Notch2* in centrally-located cells presented with DLL4 (Figure 3C), which we had previously observed in untreated cultures (Figure 2E).”

The purpose of this experiment is to determine whether TGFβ signaling impacts the formation of the gradient of *Jag1* and *Notch2* mRNA expression. The results indicate that TGFβ signaling is not required for gradient formation but nonetheless modulates the responsiveness of central cells to presentation with DLL4. This modulation of responsiveness is consistent with our prior work in liver progenitors demonstrating that TGFβ induces expression of both Notch ligand and receptor (Kaylan et al., 2016), both of which we show is important for fate segregation within patterns elsewhere in the manuscript. Throughout Figure 3, we included DLL4 as the representative Notch ligand, because it consistently induced a robust baseline pattern of differentiation for modulation with the various inhibitors in these experiments.

5) To validate the claim that YAP plays a role in progenitor segmentation, the authors should try to quantify cytoplasmic YAP localization on soft (~ 4 kPa) substrates.

We agree that in order to evaluate the potential role of YAP, a more thorough and quantitative analysis of YAP expression was required, beyond our original qualitative description. Accordingly, we quantitatively examined the expression of YAP in the arrayed islands and now have reported this spatial quantification for both 4 kPa and 30 kPa substrates in Figure 6E. Notably, consistent with biliary differentiation markers and Notch ligands/receptors, YAP exhibited an increased expression at the periphery of these cell patterns. Peripheral expression of YAP was observed for both 4 kPa and 30 kPa substrates (Figure 6E), but this pattern of YAP expression was not influenced by the presence of Notch ligands in the array domains (data not shown).